# Racial and Ethnic Disparities in European Breast Cancer Clinical Trials

**DOI:** 10.3390/cancers16091726

**Published:** 2024-04-29

**Authors:** Angelina Bania, Antonis Adamou, Emmanouil Saloustros

**Affiliations:** 1Faculty of Medicine, School of Health Sciences, University of Patras, 26504 Patras, Greece; bania.ang@gmail.com; 2Institute of Diagnostic and Interventional Neuroradiology, Hannover Medical School, 30625 Hannover, Germany; antonadamou@gmail.com; 3Division of Oncology, Faculty of Medicine, School of Health Sciences, University of Thessaly, 41110 Larissa, Greece

**Keywords:** breast cancer, clinical trials, race, ethnicity, minorities, representation, inclusion

## Abstract

**Simple Summary:**

Breast cancer is known to be associated with the race and ethnicity of patients regarding the tumor characteristics and patient survival. However, in worldwide clinical trials, the participation of Black and Hispanic patients is lower than expected based on the frequency of breast cancer in these populations. This article aims to assess race reporting and representation trends in European trials. Ninety-seven such trials conducted exclusively in Europe between 2010 and 2022 were identified in the PubMed and ClinicalTrials.gov databases. Race was reported in 10.31% of these, and mostly in trials carried out across multiple European countries. They featured a White-predominant population, with 1.08% Asian and 0.88% Black patients included. Race reporting trends in European trials are much lower than in worldwide or American-based trials on the same subject. Systematic race reporting among other patient demographics and adequate minority inclusion will ameliorate the quality of European clinical trials and promote equality in healthcare access.

**Abstract:**

Breast cancer is the most prevalent female cancer worldwide with known correlations between the race and tumor characteristics of the patients and prognosis. International and US-based studies, however, have reported a disproportionate representation of Black and Hispanic patients in clinical trials. This is the first study assessing race and ethnicity reporting trends and inclusion in European breast cancer trials. The PubMed and ClinicalTrials.gov databases were systematically searched for trials on breast cancer treatment conducted exclusively in Europe between 2010 and 2022. Of the 97 identified trials, race was reported in 10.31%. Multinational participation, but not the study size or trial phase, was significantly associated with higher race reporting trends. These 10 trials featured a White-predominant population, with 1.08% Asian and 0.88% Black patients included. The acquisition of the race and ethnicity data of patients in European trials is lower compared to the U.S. or worldwide studies and does not permit extensive analysis of minority participation. In a limited analysis, the low rates of minority participation are concerning, based on population-based data on minorities in select European countries. These observations should encourage race reporting practices in European breast cancer trials and adequate minority participation to support the generalizability of the results of the studies and promote healthcare equity.

## 1. Introduction

Breast cancer is the most common cancer in women, and the second most common in terms of absolute number of deaths [1]. Subsequently, it is a major point of focus for both pre-clinical and clinical research that has revolutionized treatment options and contributed to the tremendous improvement in breast cancer survival over the past years.

However, the improvement is not equal between populations of different racial and ethnic backgrounds. Race impacts cultural, phenotypic, and genetic diversity, and therefore alters the disease risk, biology, and health outcomes. Despite both being relevant to an individual’s background and sometimes being mistakenly used interchangeably, race and ethnicity carry different meanings and sub-categorizations. Race is usually more dependent on the physical and/or biological traits of a group, while ethnicity refers mostly to a group’s culture and ancestry. What still remains unknown is whether the different races and ethnicities are being included equally in ongoing breast cancer clinical trials.

These become relevant to breast cancer and lead to differential incidence, mortality, and treatment responses among subgroups [2]. The most prominent and well-established difference lies in the higher incidence of aggressive breast cancer subtypes in Black women, which translates to 41% higher breast cancer mortality compared to White women, although their overall incidence of breast cancer is slightly lower (127 versus 133 per 100,000) [1]. This disparity extends to male breast cancer, where Black men are more likely to develop breast cancer, both overall and in terms of aggressive subtypes [3]. The Hispanic population, on the other hand, is less likely to develop breast cancer than non-Hispanic Whites (96 per 100,000) [1]. However, given that the Hispanic ethnicity is associated with various geographical origins, European ancestry is more highly associated with increased breast cancer risks among other Hispanic genetic ancestries [4,5]. Asian women and Pacific Islanders in the USA have the second lowest incidence (99 per 100,000) and lowest mortality among the major studied groups (White, Black, Asian/Pacific Islander, American Indian, and Alaska Native and Hispanic/Latino) [1].

Patients from different racial and ethnic groups should have an equal chance and awareness of participating in clinical trials, as this is crucial for the development of standard-of-care therapies that are both effective and tolerable for patients from diverse backgrounds. The issue of racial and ethnic minorities in breast cancer trials has been addressed in multiple articles from U.S.-based researchers [6,7,8,9,10,11]. Despite the fact that minorities have a similar level of willingness to be enrolled in clinical trials compared to White patients [12], major differences seem to arise when comparing their participation in published trials. These reports examined trials conducted either exclusively in the United States or regardless of country of origin, and in most cases compared them with the U.S. cancer incidence [6,7,8,9,11] and U.S. Census Bureau data [7]. However, no analysis on the racial and ethnic diversity in European-based clinical trials for breast cancer has been published in the existing literature. In this study, we assessed the race and ethnicity reporting trends among European breast cancer drug trials as well as the representation of minorities in the trial population.

## 2. Materials and Methods

An electronic search of PubMed (MEDLINE) was performed using a predesigned search algorithm. The algorithm was written as follows:

(((breast cancer) AND (clinical trial)) AND (drug)) AND (effectiveness) AND (((((((((((((((((((((((((((((((((((((((((((((((((Europe) OR (United Kingdom)) OR (Great Britain)) OR (Ireland)) OR (Iceland)) OR (Norway)) OR (Sweden)) OR (Finland)) OR (The Netherlands)) OR (Belgium)) OR (Germany)) OR (Denmark)) OR (France)) OR (Luxemburg)) OR (Switzerland)) OR (Spain)) OR (Portugal)) OR (Italy)) OR (Poland)) OR (Latvia)) OR (Lithuania)) OR (Esthonia)) OR (Russia)) OR (Ukraine)) OR (Czechia)) OR (Czech Republic)) OR (Hungary)) OR (Greece)) OR (Slovenia)) OR (Slovakia)) OR (Croatia)) OR (Serbia)) OR (Bulgaria)) OR (Romania)) OR (Turkey)) OR (Israel)) OR (Cyprus)) OR (Malta)) OR (Austria)) OR (Montenegro)) OR (Albania)) OR (Bosnia)) OR (North Macedonia)) OR (Belarus)) OR (Kosovo)) OR (Faroe Islands)) OR (Georgia)) OR (Azerbaijan)) OR (Armenia)) OR (Europe) AND (phase 2) OR (phase 3)

Relevant studies were identified based on predetermined inclusion and exclusion criteria. Additional studies were identified using the ClinicalTrials.gov database for completed Phase 2 or 3 trials on breast cancer in all European countries. Only articles fully available in the English language were included. The last search was performed on 13 February 2023.

### 2.1. Inclusion and Exclusion Criteria

Only Phase 2 or 3 clinical trials with at least 250 participants conducted in Europe including Turkey and Israel, with results published within 2010–2022, were included in this systematic review. Single-center, multicenter, and multinational trials were eligible. The included trials examined the efficacy of pharmacological agents, drug combinations, and regimens on the treatment of breast cancer. No restriction on patient gender or age was imposed. Retrospective studies, case–control studies, case reports, and series were excluded. Non-European trials as well as multicenter trials with recruited participants from centers outside Europe were excluded. Safety studies with no mention of efficacy were also not included.

### 2.2. Data Extraction

Using a predetermined data table, the following data were extracted from all included studies: article first author, trial title, year of publication, participating countries, trial phase, sample size, and race or ethnicity, where applicable. From the studies that reported participant race, additional data were retrieved: male to female ratio, the breast cancer molecular subtype studied and the number of patients in each racial or ethnic group. Based on the official definitions mentioned in the introduction, the common practices on race reporting in medical research and the data arising in the included articles, in this review, the term race encompasses the categories White/Caucasian, Asian, Black, Native American, and Pacific Islander, while the term ethnicity includes Hispanic/Latino, Indian, East Asian, and Southeast Asian.

### 2.3. Statistical Analysis

The race and ethnicity report rate was calculated as the proportion of trials reporting race and/or ethnicity over the total number of included studies. Reporting trends were compared between single versus multi-country trials, Phase 2 versus Phase 3 trials, and small versus large trials. A cut-off of 1000 participants was employed to subcategorize the clinical trials by population size. Statistical significance was determined by the chi-squared test. Significance threshold was set at a *p*-value < 0.05.

In order to assess each racial group’s representation in European trials, the total number of participants from each group in all ten trials was calculated and expressed as a proportion of the whole sample. Given the substantial number of patients of Unknown or Other race as well as the fact that some studies included Hispanic/Latino as a racial category, the number of patients in each racial group was also expressed as a proportion of the total patients in these four groups.

To assess the adequacy of the representation of minorities in breast cancer clinical trials, the calculated percentages of each racial group were compared to its respective percentage in the general population. In a similar U.S.-based study [6], the trial-derived data were compared to the cancer incidence per race and ethnicity for the respective time period as reported in the National Cancer Institute Surveillance, Epidemiology, and End Result (SEER) database. However, no such information is available for European patients, and therefore population-wide statistics were used to provide a rough estimate.

Statistical analysis was performed using MS Excel. Figures were designed using GraphPad Prism version 8 for Windows, GraphPad Software, www.graphpad.com.

## 3. Results

The PubMed literature search yielded 1367 results, of which 233 were assessed as full-text, and 86 clinical trials were eventually included. Eleven additional studies were identified and included from ClinicalTrials.gov, thus raising the final number of included studies in this systematic review to 97 [13,14,15,16,17,18,19,20,21,22,23,24,25,26,27,28,29,30,31,32,33,34,35,36,37,38,39,40,41,42,43,44,45,46,47,48,49,50,51,52,53,54,55,56,57,58,59,60,61,62,63,64,65,66,67,68,69,70,71,72,73,74,75,76,77,78,79,80,81,82,83,84,85,86,87,88,89,90,91,92,93,94,95,96,97,98,99,100,101,102,103,104,105,106,107,108,109] (Figure 1).

The trials were published between 2010 and 2022 and included a total of 113,045 participating patients. Seventy trials were conducted within a single country. Of these, eighteen were conducted in Germany, sixteen in Italy, seven in the United Kingdom, six in the Netherlands, five in France and Greece, respectively, four in Denmark, three in each of Austria and Spain, two in Sweden, and one in Finland. Twenty-seven trials were multinational, comprised of trial centers within two to fourteen countries. One article referred only to the Italian subpopulation of an international trial with both European and non-European centers and was included in this review. Among the trials clearly stating the study phase, 2/14 were Phase 2 trials, and 8/80 were Phase 3 trials.

### 3.1. Race and Ethnicity Reporting in European Breast Cancer Clinical Trials

First, we assessed whether race and ethnicity were reported in the demographics section of the included European trials. Out of the 97 included trials, race or ethnicity data were collected and presented in 10 trials [46,47,48,49,64,67,69,100,106,109] (10.3%), or for 12,179 out of a total of 113,045 patients (10.77%).

Among these ten trials, four focused on hormone receptor-positive, human epidermal growth factor receptor 2-negative breast cancer (HR+, HER2-), one focused on HER2-positive breast cancer only, two studies recruited patients of multiple breast cancer subtypes, and three studies made no mention of the patients’ subtype.

Three out of ten trials made a distinction between race (White/Caucasian, Black, Asian, Native American) and ethnicity (Hispanic/Latino, Indian, East Asian, Southeast Asian). Two of them reported both entities separately, and one trial reported to have collected data only on race but not ethnicity. Therefore, it should be noted that in these two studies [69,100], the Hispanic/Latino ethnic group may have partially overlapped with the White/Caucasian and possibly the Black racial groups. It is also possible that in the trial by Jerusalem et al. [100], the term Hispanic/Latino refers to the Spanish population, instead of the population originating from Latin America, due to the large number of Hispanic/Latino enrolment. In the remaining seven studies, the terms ethnic origin, ethnicity, and race were all used to encompass the categorizations White/Caucasian, Black, Asian, and Hispanic/Latino.

A rough correlation between study size, phase, and participating countries and race reporting trends was attempted. Two out of the ten studies that reported race or ethnicity were Phase 2 trials, while the remaining eight were Phase 3. It is statistically unlikely for there to be an association between race reporting and phase (*p*-value = 0.64), as it was mentioned in two out of fourteen (14%) Phase 2 and eight out of eighty (10%) Phase 3 trials (Figure 2A).

Using a cut-off of 1000 participants, 10 trials that had provided a racial or ethnic breakdown of their sample were characterized as large (3/10, 30%) or small (7/10, 70%). When expressed as a proportion of all included studies, three out of forty-two (7.1%) trials with more than 1000 participants reported their race or ethnicity compared to seven out of fifty-five (12.7%) smaller studies with less than 1000 recruited patients (*p*-value = 0.40) (Figure 2B).

Six out of ten (60%) trials with racial or ethnic breakdown were multinational. This corresponded to six out of twenty-seven (22%) multinational trials reporting race compared to four out of seventy (5.7%) single-country trials. If the fact that one of these studies contained the Italian subpopulation of a larger international trial is considered, the proportional difference in race reporting between international and single-country trials rises further (7/28 or 25% versus 3/69 or 4.3%, *p*-value = 0.004) (Figure 2C).

Interestingly, only three [22,23,86] studies other than the ten trials identified initially, acknowledged the lack of race or ethnicity data collection or referred to it as a limitation of their study.

### 3.2. Inclusion of Racial and Ethnic Minorities in European Breast Cancer Clinical Trials

Second, we became interested in studying the representation of different ethnic and racial groups in these 10 trials. Only groups appearing in at least two trials were analyzed. These included White/Caucasian (ten trials), Asian (seven trials), Black (four trials) and Native American (two trials) races and the Hispanic/Latino ethnicity (four trials). The Pacific Islander race and the Indian, East Asian, and Southeast Asian ethnicities were only mentioned once and included a minimal number of participants, thus not being discussed further.

Hence, from the total of 12,179 patients in these 10 trials, 11,284 (92.65%) patients were White/Caucasian, 132 (1.08%) were Asian, 107 (0.88%) Black, 6 (0.05%) were Native American, and the remaining were classified as Other/Unknown. A total of 11,529 patients identified as either White/Caucasian, Asian, Black, or Native American. The relative percentages for these groups were 97.9% for White/Caucasian, 1.14% for Asian, 0.93% for Black, and 0.05% for Native Americans.

The Hispanic/Latino ethnicity was the one most widely mentioned in the studies and accounted for 390 out of 12,179 (3.20%) participants.

Among the patients participating in trials focusing exclusively on hormone receptor-positive, human epidermal growth factor receptor 2-negative breast cancer (HR+, HER2-) breast cancer, Caucasian predominance was even more remarkable, as these four trials featured a 97.4% White, 0.26% Asian, and 0.26% Black population among the 3454 patients.

### 3.3. Representation of Racial Minorities in Breast Cancer Clinical Trials

The White, Asian, and Black racial groups were examined in terms of representation in single-country clinical trials relative to their proportion in the country-specific general population. No data on the presence of Native American people in Europe were available for comparison. An assessment of the Hispanic/Latino population representation was also not possible due to the low availability of data for comparison as well as the differences in the definition of this group.

Race is not generally recorded in the population statistics of most European countries, except for the United Kingdom. According to the 2021 Census of England and Wales [110], 81.7% of the population was White, followed by Asian (9.3%), Black (4.0%), Mixed (2.9%), and Other (2.1%). In the PERSEPHONE [109] trial conducted in 152 UK hospitals and published in 2019, 3306 were White (93.1%), 109 were Asian (3.07%), 97 were Black (2.73%), and 38 (1.07%) belonged to other races (excluding 538 patients of Unknown race).

Two [64,69] out of the ten studies pertained to Italian breast cancer patients. The Italian Permanent Census of Population and Housing (Censimento permanente della Popolazione e delle Abitazioni) does not provide data on the racial distribution of the Italian population but provides data on the national background of foreign residents. At the beginning of 2022, Asian and African residents were calculated to account for about 1.91% (1,126,582 out of 59,030,133) and 1.92% (1,135,756 out of 59,030,133) of the population, respectively [111,112]. Given that Black Italian citizens and Italian citizens of Asian origin were not accounted for in the aforementioned calculated percentages, they are probably underestimations. Two of the 10 studies pertained to Italian breast cancer patients. After excluding patients of unknown race, the two studies included 976 participants, three of which were Asian (0.31%). No Black patients participated in either of these trials.

## 4. Discussion

In this study, we report an alarmingly low frequency of race reporting in European breast cancer drug clinical trials (10.3%), especially in single-country trials (4.3%). In comparison, in U.S. American trials, racial breakdown was provided in 66.67% (8 out of 12) of trials leading to breast cancer oral chemotherapy FDA approval [11] and 55% (38 out of 69) in breast cancer precision oncology trials [6]. Among the breast cancer immunotherapy trials regardless of country of patient enrolment, 79% (19/24) of the trials provided a racial breakdown of the participants [9]. Our exclusive European selection of 97 trials lies far behind, with five to almost eight times lower race and ethnicity reporting rates.

The issue of race/ethnicity reporting in U.S. trials, regardless of the disease studied, has already been raised in the literature, with various authors reporting rising trends in reporting, in part attributed to the requirements set by the National Institute for Health (NIH), the U.S. Food and Drug Administration (FDA), and ClinicalTrials.gov [113,114,115,116,117]. The difference in race reporting between the U.S. and Europe can be due to differences in the regulatory guidelines, since no such policies exist in Europe [117].

Another crucial issue is the consistency of race reporting among trials, and it would be optimal to adopt a uniform data collection methodology and racial/ethnicity terminology. In the present study, we suspect that the terms ethnic origin, ethnicity, and race have been given overlapping definitions in various trials and thus used interchangeably, while about 5% of patients were categorized as Unknown. A similar trend was observed in a report by Candelario et al. [117], which was attributed to the lack of familiarity of Europeans with the various race/ethnicity terms compared to the USA, where race data collection is much more commonplace.

Our limited analysis of the 12,179 patients for which race/ethnicity data were available hints toward an underrepresentation of Black and Asian patients, even compared with the population demographics of the respective countries. According to Aldrighetti et al. [6], Black and Hispanic patients are underrepresented, while White and Asian patients are overrepresented in breast precision oncology trials, when comparing their enrolment rates with the expected enrolment based on the proportion of each group in the U.S. breast cancer patient population. Similarly, Black and Hispanic patients were underrepresented in breast therapeutic oncology trials compared to Asian and White patients [7]. In breast immunotherapy trials, Asian and Black patients were underrepresented, with a 2.5-fold and 11-fold lower enrolment, respectively, compared to the CDC-age adjusted incidence [9]. Black patients were also specifically found to be underrepresented in breast oncology trials leading to FDA drug approval [8,10,11].

On the other side of the Atlantic, the United States Census Bureau [118], in accordance with the Office of Management and Budget standards, has recognized five major racial groups: White, Black or African American, American Indian or Alaska Native, Asian, Native Hawaiian or Other Pacific Islander. People may choose to self-identify as multiple races. Similarly, the 2021 Census of England and Wales [119] defined the following ethnic groups: White, Black or Black British, Caribbean or African, Asian or Asian British, Mixed and Other ethnic groups. In terms of ethnicity, the U.S. Census Bureau divides the population into Hispanic/Latino and non-Hispanic/Latino [120].

In Europe, on the other hand, there are scarce resources regarding race and ethnicity, and reports are derived from each individual country instead of the European Union as a whole. This makes the investigation of possible discrepancies in race and ethnicity representation in European breast cancer clinical trials challenging. As previously mentioned, the definition of ethnicity and its components varied among the included trials. There was no distinction between White and non-White Hispanics, and it is suspected that this group is highly heterogenous, and in some cases mostly includes people of Spanish nationality. It is, however, unclear whether all trials with recruited Spanish patients categorized them in the Hispanic/Latino ethnic group.

In contrast to the U.S., where breast cancer prevalence and mortality by race is reported and available on the National Cancer Institute Surveillance, Epidemiology, and End Result (SEER) database, no similar registry exists for Europe. The unavailability of such data is hindering our efforts to assess the representation of minorities in the identified trials with respect to the race-specific prevalence of breast cancer per country. Interestingly, the European Statistical Office (Eurostat) does not collect race and ethnicity data among other demographic information on the European population either, based on the European Union’s policy of non-discrimination. This shows that although race and ethnicity information is relevant to breast cancer personalized care and optimal trial design, cultural considerations may be the reason behind the tendency of European breast cancer trials to underreport them. This hypothesis can be further supported by the fact that some European societies such as Sweden reject the racial categorization employed in the USA as outdated, unscientific, and even offensive [121]. Another study has raised concerns regarding a potential harm to patients from inquiring their ethnicity, since vulnerable populations are in fear of experiencing treatment based on stereotypes and receiving inferior care [122]. Therefore, any potential effort to expand the race/ethnicity reporting regulations to European trials should only be performed with careful consideration of each country’s distinct societal norms and attitudes. In the clinical setting, patient confidentiality, freedom, and equality are non-negotiable.

Based on data from the European Commission [123], the largest ethnic minority in Europe is the Roma, an umbrella-term used to describe a variety of populations (Roma, Sinti, Kale, Romanichels, Boyash/Rudari, Ashkali, Egyptians, Yenish, Dom, Lom, Rom and Abdal, and other Traveler populations). Although the ethnicity reporting system employed in our included studies relied on other sub-categorizations, the Roma were not once mentioned in any of the 97 trials. Therefore, although no definite conclusions can be safely drawn for this large but marginalized European group, the lack of any mention raises the suspicion of their potential underrepresentation in the trials. However, it is possible that these people were classified among the existing categories, similarly to the 2021 Census of England and Wales, which recently considered them as part of the White ethnic group.

Even though Europe is a relatively ethnically homogenous continent, we consider these findings to be concerning, especially in recent years, where large immigration waves in Europe are transforming its demographics. Racial and ethnic background can affect clinical trial results, where Black patients experience worse outcomes than White patients [124] and drugs have been shown to demonstrate inferior efficacy and more adverse effects [125]. All of these could hint at potential differences in the tumor biology in patients of different ancestries [125,126].

In any case, an inadequate recruitment of patients from racial and ethnic minorities in clinical trials limits opportunities for subgroup analyses of trial results and can lead to failure to identify differential responses to treatment based on background. In a wider perspective, a lack of diversity poses the risk of selection bias, challenges the accuracy of treatment efficacy assessment, and reduces the generalizability of European trials.

The underrepresentation of minorities in trials is unfortunately a more generalized phenomenon that is not unique to breast cancer or Europe. In 1993, the U.S. Congress passed the National Institute of Health (NIH) Revitalization Act requiring all NIH funded clinical research studies to appropriately recruit minorities. Since then, however, progress in the USA has been minimal [127], possibly hinting that strict regulations alone are inadequate to resolve this complex social and medical issue. A lack of information and understanding and the presence of mistrust and fear of clinical trials were the major deterring factors against trial participation in a small survey of African American cancer survivors [128], while healthcare professional bias toward minorities and financial and social injustice also played a role. The AACR Cancer Disparities Progress Report [2] suggests patient education within their communities, patient navigation by healthcare workers, and the selection of hospitals serving minorities as clinical trial centers as potential measures to increase minority participation. Given the differences between European and American society as well as the structure of their respective healthcare systems, identifying the specific barriers of underrepresented European patients via surveys must be a first step in addressing them.

Numerous factors mostly pertaining to the nature of our study topic and the studied clinical trials themselves should be taken into account in the interpretation of our results. First, Europe consists of multiple countries, some of which are very racially homogenous. Second, the racial categorization employed in the USA is not common in Europe. A characteristic example encountered in this study was the possibility that the Spanish population had been classified as Hispanic, which in the United States mostly refers to people originating from Central and South America. Moreover, other ethnic groups such as the Romani are more relevant to Europe than to the USA.

Our attempt to assess the adequacy of minority representation was limited by the low race/ethnicity reporting rates among trials and the lack of a European registry that includes the racial categorization of breast cancer patients. Comparisons were thus attempted by using the population-wide demographics for the selected countries only, as no such data are available for Europe as a whole. Thus, the registration of race and ethnicity of the European population by a central authority, namely the European Union via the European Commission, and the implementation of well-designed clinical trials reporting data on the race and ethnicity of the included subjects is encouraged.

## 5. Conclusions

To our knowledge, this is the first study assessing the representation of minorities in breast cancer trials in Europe. Although further research is required, we express our concern regarding the low rates of race and ethnicity reporting in European breast cancer trials, which does not allow for the extraction of safe conclusions on minority representation. Therefore, the European cancer trial community should take this into account and ensure equity and diversity in clinical trials. Finally, since race and ethnicity are self-reported by patients, inquiries of sensitive personal information should be made with discretion and respect.

## Figures and Tables

**Figure 1 cancers-16-01726-f001:**
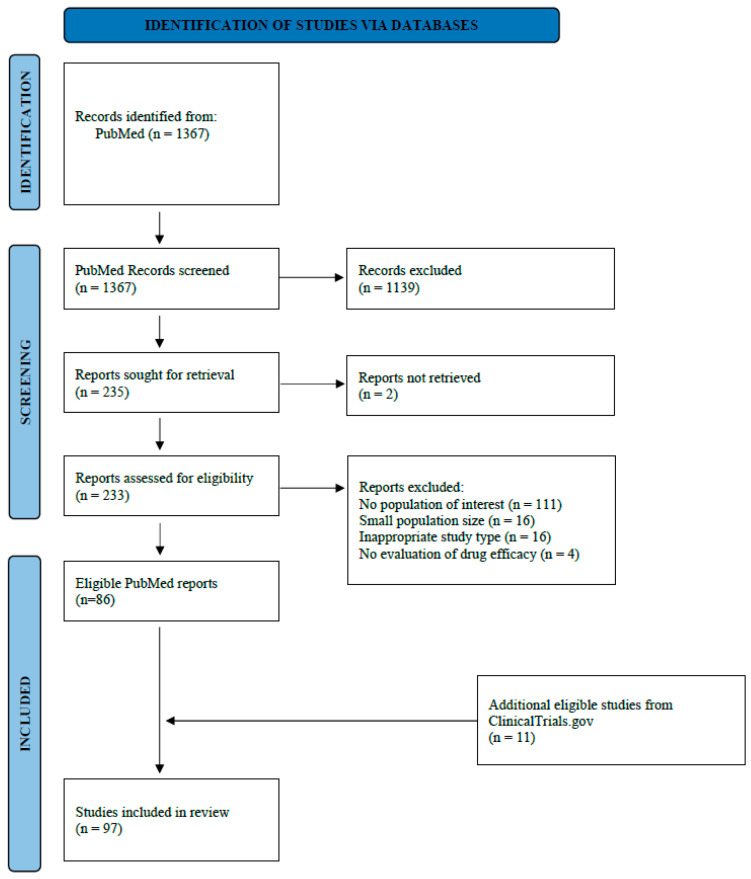
Study inclusion flow diagram.

**Figure 2 cancers-16-01726-f002:**
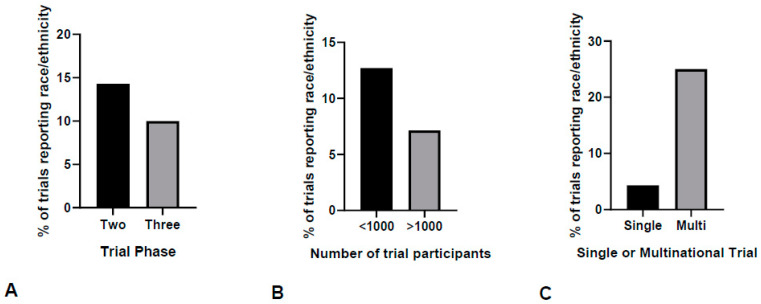
Race reporting trends according to the study characteristics. The association of race reporting and the three study characteristics of (**A**) trial phase, (**B**) study size with a cut-off of 1000, (**C**) single-country or multinational status of trial was assessed. (**A**) Race reporting rates were similar between the Phase 2 and Phase 3 trials (14% vs. 10%, *p* = 0.64) and (**B**) between small and large trials (12.7% vs. 7.1%, *p* = 0.40). (**C**) Race reporting rates were higher in multinational compared to single-country trials (25% vs. 4.3%, *p* = 0.004).

## Data Availability

Data are available upon request.

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
