# Peer review of "Racial and Ethnic Disparities in European Breast Cancer Clinical Trials"

_cancers, 2024, doi:10.3390/cancers16091726_

Round 1

Reviewer 1 Report

Comments and Suggestions for Authors

Thank you for the opportunity to review this manuscript. The manuscript “Racial and ethnic disparities in European breast cancer clinical trials” presents an important topic. The study is well conducted and presented. I have only couple of suggestions for further improvement as mentioned below:

 The authors state and conclude low rates of race and ethnicity reporting in European breast cancer trials. However, the statement is not well described and discussed. One would expect detailed discussion (reasons, rationale) on one of the main findings of the study (in the discussion section).

   The authors have described and presented search terms used for the extraction. However, to be sure that there has not been any study conducted before, it will be good to present the complete syntax with Boolean operators used (Search algorithm) to search the database in the main file rather than the supplementary file.

Comments on the Quality of English Language

Minor editing required

Author Response

We would like to thank the distinguished reviewer for taking the time to read our manuscript and provide helpful recommendations. We are glad that the reviewer recognizes the scientific merit of our article.

Reply to comment 1: Thank you for this insightful comment. We have enhanced our Discussion section with more potential reasons for the lower race reporting trends in Europe compared to the USA.

Reply to comment 2: Thank you for this suggestion. We are glad that you took interest in the complete syntax of the search algorithm and we added it into the Methods section of the manuscript.

Reviewer 2 Report

Comments and Suggestions for Authors

I thank the authors of the manuscript "Racial and Ethnic disparities in European Breast Cancer Clinical Trials" for their work and for drafting the text. 

The manuscript is well-written, clear and flowing.

Figure 2 and Figure 4 appear superfluous, they have no real reason to be present in the manuscript. I advise the authors to remove such figures. Write in the text as shown in the figure.

I ask the authors to correct the following errors in the text of their manuscript.

Line 117: p-value <0.05.

Author Response

We would like to thank the reviewer for taking the time to read our manuscript and provide helpful recommendations. We are glad that the reviewer recognizes the scientific merit of our article.

Reply to comment 1: Thank you for this suggestion. Indeed, these figures serve for visualization, rather than provide information and can be omitted.

Reply to comment 2: Thank you for identifying this, it has been corrected.

Reviewer 3 Report

Comments and Suggestions for Authors

Overall, your study makes a valuable contribution to the literature on healthcare disparities and clinical trial diversity. With some minor revisions and additions, the manuscript will be well-positioned to inform future research and practice in this important area.

1. The statement " 2.3 Statistical Analysis, However, no such information is available for European patients, and therefore population-wide statistics were used to provide a rough estimate" raises a critical methodological concern. Relying solely on population-wide statistics to estimate the representation of racial and ethnic minorities in European breast cancer trials introduces a significant limitation to the study's validity and generalizability. This approach overlooks the heterogeneity of racial and ethnic demographics across European countries and regions, as well as potential disparities in healthcare access and participation in clinical trials among different populations.

To address this limitation, the authors should consider alternative strategies for estimating the representation of minorities in European breast cancer trials

2. The statement “ 5. Conclusions. since race and ethnicity are self-reported by patients, inquiry of sensitive personal information should be made 348 with discretion and respect."

How can researchers ensure the respectful and discreet collection of race and ethnicity data, considering that such information is self-reported by patients?

3. Are there any clinical trial reports specifically categorized by breast cancer subtype(basal, luminalA/B, Her2)?

Comments on the Quality of English Language

The manuscript effectively communicates the study's findings and recommendations. With some minor improvements in language quality, it can further enhance readability and impact.

Author Response

We would like to thank the reviewer for taking the time to read our manuscript and provide valuable recommendations. We are glad that the reviewer recognizes the significance of our article.

1. Thank you for this interesting comment, which has indeed been a great cause of concern among the authors. Despite our efforts to retrieve data on the race-specific prevalence of breast cancer in Europe, no such registry exists (lines 307-314 of the revised manuscript) and we acknowledge this as a limitation in assessing the adequacy of minority representation in the trials. The demographic heterogeneity among countries has also been taken into consideration (line 365-380 in revised manuscript) and therefore we only made limited country-specific comparisons for the UK and Italy, where some population-wide statistics were available online. Indeed, one cannot draw conclusions for Europe as a whole based on the available data.

2. Thank you for this comment, as this is indeed worth clarifying. In lines 307-325 of the Discussion section of our revised manuscript, it was identified that Eurostat avoids collecting race/ethnicity data of the European population to avoid the risk of discrimination. This shows that race/ethnicity might be a sensitive topic for European citizens, compared to American citizens where race self-reporting is commonplace and obligatory in many formal documents. Our comment to use discretion and respect when requesting the race or ethnic status of a patient aims to highlight the sensitive nature of this information, the need for confidentiality and serve as a reminder that a polite, non-discriminatory attitude is necessary in all patient encounters. We now added this information in the Discussion section of our manuscript.

3. Thank you for this insightful comment. Indeed it would be interesting to compare the inclusion of minorities among trials focusing on specific breast cancer subtypes, given the different prevalence of subtypes among racial groups. Amont the trials reporting patient race, we identified 4 focusing on HR+, HER2- cancer only and 1 on HER2+ cancer only. We had calculated the respective minority participation rates according to breast cancer subtype studied and now included these data in our manuscript. However, given the very limited number of studies making this distinction the data was inadequate to perform comparisons between groups.